# Macrobiological Degradation of Esterified Wood with Sorbitol and Citric Acid

**Andreas Treu** [1,*] **, Lina Nunes** [2,3] **and Erik Larnøy** [1]

1 Norwegian Institute of Bioeconomy Research, Høgskoleveien 8, 1433 Ås, Norway; erik.larnoy@nibio.no
2 Structures Department, LNEC, National Laboratory for Civil Engineering, Av. do Brasil, 101, 1700-066 Lisbon, Portugal; linanunes@lnec.pt
3 Centre for Ecology, Evolution and Environmental Changes (cE3c)/Azorean Biodiversity Group and University of Azores, Rua Capitão João d'Ávila, 9700-042 Angra do Heroísmo, Portugal
* Correspondence: andreas.treu@nibio.no; Tel.: +47-45671343

**Abstract:** There is a need for new solutions in wood protection against marine wood borers and termites in Europe. A new solution could be the esterification of wood with sorbitol and citric acid (SCA) since these are inexpensive and readily available feedstock chemicals and have shown protective properties against fungal wood degradation in earlier studies and prevented macrobiological degradation, as shown in this study. Protection of wood products in the marine environment lacks available wood preservatives that are approved for marine applications. Termite infestation is opposed mainly by biocide treatments of wood. Several wood modification systems show high resistance against both marine borers and subterranean termites. However, the existing commercialized wood modification products are costly. Both macrobiological forms of degradation represent a great threat for most European wood species, which are rapidly and severely degraded if not properly treated. This study investigated esterified wood in standard field trials against marine wood borers, and against subterranean termites in laboratory trials in a no-choice and choice test. The treatment showed good resistance against wood borers in the marine environment after one season and against subterranean termites in the laboratory after eight weeks. The low termite survival rate (SR) in the no-choice test during the first week of testing indicates a mode of action that is incomparable to other wood modification treatments.

**Keywords:** subterranean termites; marine wood borers; polyesterification; wood protection

## 1. Introduction

Biodeterioration of most untreated European wood species in the marine environment and in termite-infested areas can be very rapid and severe. However, attack by these organisms is limited by the geographical distribution of the respective species. Shipworms ingest wood particles produced by the grinding action of their shells, while termites manipulate lignocellulosic material with their mouth parts. Even though their attack pattern affects the macroanatomical structure of wood, both organisms rely on diverse mechanisms involving multiple complementary enzymes to degrade lignocellulose [1,2].

The tolerance level against termites, which cause significant damage to buildings, is close to zero in countries with a relatively high standard of living. In contrast, in less-developed countries no or little action is undertaken due to the immense costs of pest management. Current pest management programs for subterranean termites may include soil or wall barrier treatments (e.g., fipronil), wood treatments and population control using baits (e.g., hexaflumuron or diflubenzuron) together with the use of physical or physical–chemical barriers (steel mesh or treated plastic sheets) and adequate design. Nevertheless, all integrated termite control strategies still rely strongly on wood or barrier treatments.

Inexpensive insecticides have increasingly disappeared due to public health concerns, leading to a smaller number of compounds available for use in pest control. Preventive treatment of wood against termite attack is dominated by the use of biocides, either pressure-treated with copper-based products for higher risk applications (e.g., Use Class 4, [3]) or insecticides like permethrin and borates. Several techniques of wood modification have also been shown to provide improved resistance to subterranean termite attack [4].

Six subterranean termites species have been described in Europe, all belonging to the genus *Reticulitermes* (Blattodea, Isoptera, Rhinotermitidae) [5], namely five native species (*R. grassei*, *R. banyulensis*, *R. urbis*, *R. lucifugus*, and *R. balkanensis*) and one introduced species (*R. flavipes*). The subterranean termite species *Reticulitermes grassei* from southwestern Europe has also been established in Britain [6] and in one of the islands of the Azores [7]. *Reticulitermes flavipes*, which is a common termite species in the eastern part of the United States, has spread to various regions of the world such as Canada, the Bahamas, Europe (well established in France and present in Hamburg, Germany), and South America [8,9].

Subterranean termites build tunnels, which connect the nest with woody resources discovered within the territory of each colony [10]. In laboratory trials, however, the food source is easily discovered by termites. The termites use the moist sand from the test container to build mud tunnels to the wood samples.

Wood structures in the marine environment are usually protected by wood preservatives, by plastic wrapping or coating [11]. However, no wood preservative is approved for marine applications in Europe. New attempts to protect wood in the marine environment in Europe include mainly the prevention of the settlement of wood borer larvae by mechanical barriers and wood modification treatments [12]. Wood modification methods proven to perform well in the marine environment for many years are acetylation and furfurylation [13]. In addition, 1.3-dimethylol-4.5-dihydroxyethyleneurea (DMDHEU)-treated wood has shown promising results after almost 10 years of exposure [4,14].

Marine wood borers include Bivalvia (Teredinidae and Xylophagaidae), Isopoda (Limnoriidae and Sphaeromatidae), and Amphipoda (Cheluridae). In Europe, most wood-boring bivalves belong to the Teredinidae, but species of the Xylophagaidae, such as *Xylophaga dorsalis*, have also been reported as present near the sea bed in Europe [15]. Comprehensive studies on the biogeography of wood-boring crustaceans (Limnoriidae) and bivalve wood borers (Teredinidae) in European coastal waters have shown the diversity of established species and their past and recent distribution [16,17].

Wood modification provides dimensional stability and extends the service life of wood by protecting it from biological attack without the use of biocides. The main commercialized wood modification processes have gained market share in Europe across a wide range of use classes. However, none of the commercialized wood modification systems are certified for use in termite-infested areas or the marine environment. Modified wood is gaining more favor with architects, specifiers, and end-users, which suggests a continued success [18]. Nevertheless, the commercialized wood modification systems are significantly more expensive than traditional alternatives containing biocides and can still be considered a niche product. Thus, a new, low-cost, and non-toxic wood modification system is urgently needed. Inexpensive and readily available feedstock chemicals, which are more environmentally benign, have been used in the treatment of wood with a combination of sorbitol and citric acid (SCA modification). Earlier studies on the utilization of sorbitol for wood modification have been limited to describing solely the dimensional stabilization of the wood matrix [19]. In recent studies, SCA-modified wood was shown to exhibit not only enhanced dimensional stability, but also durability against decay fungi and reduced susceptibility to blue stain fungi [20].

Several studies have used citric acid and glycerol for wood protection. Essoua et al. [21] and L'Hostis et al. [22] observed, among other properties, improved mechanical properties and better protection against fungal decay. Berube et al. [23] found that the use of acid catalysts resulted in higher levels of polymerization. Guo et al. [24] investigated activated glucose and citric acid using Scots

pine sapwood. Citric acid has also been used as a wood modification method to improve the water resistance of the end grain of butt joints in wood welding [25] and as a wood modification agent for solid wood, where it acts as a cell wall crosslinking agent [26,27].

The resistance of SCA-treated wood against wood borers in the marine environment and against subterranean termites has not been studied earlier and is presented in this study. The aim of this study was to expose different treatment levels of SCA-treated wood using European wood species to marine wood borers in two field test sites and to a no-choice (forced feeding) test and a two-choice test using subterranean termites in laboratory trials.

## 2. Materials and Methods

### 2.1. Wood Treatments

Scots pine sapwood (*Pinus sylvestris* L.) and spruce wood samples (*Picea abies (L.) H.Karst.*) were treated using different concentrations of a sorbitol and citric acid (SCA) stock solution. The stock solution (pH 2, density = 1.28 $g/cm^3$) was prepared from powdered citric acid (VWR Chemicals, Kokstad, Norway, CAS 77-92-9) and D-Sorbitol (Ecogreen Oleochemicals GmbH, Dessau-Roßlau, Germany, CAS 50-70-4) in a 3:1 molar ratio. The solids were dissolved in deionized water at room temperature to make a stock solution of 56% *w/w* concentration. The stock solution was then diluted to different concentrations leading to the different weight percentage gains (WPGs) of wood samples listed in Table 1.

**Table 1.** Overview of weight percentage gain (WPG) of sorbitol and citric acid (SCA)-treated Scots pine and spruce wood samples used in field tests in the marine environment and in termite exposure. SD, standard deviation.

| WPG Group | Wood Species | Average WPG (%) | SD | Type of Test |
|---|---|---|---|---|
| SCA WPG25% | *Picea abies* | 26.8 | 6.0 | |
| SCA WPG25% | *Pinus sylvestris* sapwood | 25.1 | 0.8 | |
| SCA WPG34% | *P. sylvestris* sapwood | 33.8 | 3.6 | |
| SCA WPG38% | *P. abies* | 37.9 | 10.6 | |
| SCA WPG50% | *P. abies* | 56.7 | 17.7 | |
| SCA WPG50% | *P. sylvestris* sapwood | 52.9 | 1.2 | Marine application |
| SCA WPG70% | *P. abies* | 74.4 | 7.4 | |
| SCA WPG70% | *P. sylvestris* sapwood | 70.2 | 3.5 | |
| SCA WPG84% | *P. sylvestris* sapwood | 83.9 | 18.0 | |
| SCA WPG90% | *P. abies* | 92.9 | 12.9 | |
| SCA WPG90% | *P. sylvestris* sapwood | 88.9 | 13.6 | |
| SCA WPG25% | *P. sylvestris* sapwood | 25.1 | 3.0 | |
| SCA WPG70% | *P. sylvestris* sapwood | 71.6 | 3.9 | Termite exposure |
| SCA WPG100% | *P. sylvestris* sapwood | 100.4 | 4.1 | |

Scots pine sapwood samples were treated by precipitation of calcium oxalate CaOx in the wood and the reaction by-product potassium chloride (KCl). The treatment was initiated by the reactants potassium oxalate monohydrate (KOx, Sigma-Aldrich (Merck KGaA), Darmstadt, Germany, CAS 6487-48-5) and calcium chloride hexahydrate (CaCl2, Sigma-Aldrich, CAS 7774-34-7).

Acetylated radiata pine (*Pinus radiata* D. Don) was provided by Accsys Technologies (Accoya®, Arnhem, The Netherlands).

Thermally modified ash (*Fraxinus excelsior* L.) was purchased at Maxbo hardware store and produced by Moelven, Moelv, Norway.

### 2.2. Termite Exposure

The species *Reticulitermes grassei* Clément (Blattodea: Isoptera: Rhinotermitidae) was used for the termite trials in the laboratory. The termites were collected in a forest of maritime pine (*Pinus pinaster* Ait.) in Portugal. The termite test was performed according to EN 117 [28] as a no-choice (forced feeding) test and adapted as a two-choice test with five replicates of three different concentrations of SCA, resulting in a weight percent gain (WPG) of approximately 25%, 70%, and 100% of SCA-treated Scots pine wood specimens. The wood samples with dimensions of $15 \times 25 \times 50$ mm were not leached prior to termite exposure. Glass (EN 117) or plastic (choice test) containers filled with a layer of about 6 cm loose fill and humidified sand (4 parts of Fontainebleau sand® to 1 of distilled water) were used for the tests. To each container, 250 workers were added as well as 1–3 soldiers and 3–5 nymphs. After installation of the termites, the test specimens were placed inside the containers. All test containers were kept in a conditioned chamber at $24 \pm 2$ °C and relative humidity of $80\% \pm 5\%$ for 8 weeks or until all termites were visibly dead.

Untreated sapwood samples of Scots pine and maritime pine (5 replicates) were used as controls in the no-choice test, while Scots pine was used as reference in the two-choice test (3 replicates, 6 test specimens). Termite resistance was determined according to EN 117 [28] criteria after 8 weeks of termite exposure by registering the survival rate (SR) (percentage of living termites at the end of the test) and visual examination and grading of the test specimens using the EN 117 [28] rating system from 0–4 (where 0 = no attack, 1 = attempted attack, 2 = slight attack, 3 = average attack, and 4 = strong attack, Figure S1) and additionally by wood mass loss (%).

The mass loss was calculated as follows:

$$ML\ (\%) = \frac{(m_{01} - m_{02})}{m_{01}} \times 100 \tag{1}$$

where *ML* is the mass loss after termite exposure, $m_{01}$ is the dry mass of treated or untreated control wood samples before termite exposure, and $m_{02}$ is the dry mass of treated or untreated wood samples after termite exposure. The wood dry mass was determined by drying the wood samples in an oven at 103 °C and included the mass of the SCA polymer in treated samples.

Wood moisture content was also determined after the termite exposure and calculated as follows:

$$MC\ (\%) = \frac{(m_u - m_{02})}{m_{02}} \times 100 \tag{2}$$

where *MC* is the wood moisture content after termite exposure, $m_u$ is the mass of treated or untreated wood samples after termite exposure, and $m_{02}$ is the dry mass of treated or untreated wood samples after termite exposure.

In addition, the number of termites was counted before and after testing and a survival rate (%) was calculated as follows:

$$SR\ (\%) = \frac{(N_{t1} - N_{t2})}{N_{t1}} \times 100 \tag{3}$$

where *SR* is the termite survival rate after termite exposure, $N_{t1}$ is the number of termite workers at the beginning of the termite test, and $N_{t2}$ is the number of termite workers after termite exposure.

### 2.3. Exposure to the Marine Environment

Sorbitol and citric acid (SCA)-treated wood, Norway spruce, and Scots pine sapwood, with different treatment levels (Table 1), were tested against wood borers according to EN 275 [29] at two test sites in the Oslofjord (Figure S2) during one season (May to September 2019). Norway spruce was used in addition to pine since earlier trials at the Norwegian Institute of Bioeconomy Research (NIBIO) showed that spruce was treatable with SCA. The dimension of the wood samples was $25 \times 75 \times 200$ mm.

Acetylated radiata pine samples from commercial production were used as wood modification reference as one of the known durable modification treatments, however not certified for Use Class 5, performing well in the marine environment [13]. Thermal treatment was used as an additional wood modification reference since it is a treatment resulting in increased hardness of the surface [30] and has shown fungal decay resistance in Service Class 3 (outside applications) [31].

Calcium oxalate was used as a wood treatment, since mineralization is described as resulting in an insoluble salt calcium oxalate [32]. However, the method has not been used in the marine environment before, but it was assumed that the treatment would result in increased hardness of the surface, thereby preventing teredinid larvae settlement.

Several untreated wood species were used as control samples, both softwoods and hardwoods (Table 2), since they reflect different durability, density, chemical composition, and anatomical wood structure. The resistance against wood borers was described by a rating system from 0–4 using X-ray photos from each wood sample (Figure S3).

**Table 2.** Overview of treatments tested in the marine environment.

| Short Name | Treatment | Wood Species | Test Site | Number of Samples per Treatment Level |
|---|---|---|---|---|
| SCA | Modification with citric acid and sorbitol, different concentrations (see Table 1) | *P. abies, P. sylvestris* (sapwood) | Drøbak, Moss harbor | 3 |
| Acetylation | Modification with acetic anhydride | *Pinus radiata* | Moss harbor | 6 |
| CA oxalate | Precipitation of calcium oxalate CaOx in the wood and the reaction by-product potassium chloride (KCl). The treatment is initiated by the reactants potassium oxalate monohydrate (KOx) and calcium chloride hexahydrate (CaCl2) | *P. sylvestris* (sapwood) | Drøbak, Moss harbor | 3 |
| Thermal modification | Thermal modification of ash wood | *Fraxinus excelsior* | Drøbak, Moss harbor | 3 and 6 |
| Spruce | Untreated | *P. abies* | Drøbak, Moss harbor | 3 and 6 |
| Scots pine sapwood | Untreated | *P. abies* | Drøbak, Moss harbor | 6 |
| Sessile oak | Untreated | *Quercus petraea* (Matt.) | Drøbak, Moss harbor | 3 and 6 |
| Sipo | Untreated | *Entandrophragma utile* (Dawe & Sprague) | Moss harbor | 6 |
| Beech | Untreated | *Fagus sylvatica* L. | Moss harbor | 6 |
| Radiata pine | Untreated | *P. radiata* | Moss harbor | 6 |
| Aspen | Untreated | *Populus tremula* L. | Moss harbor | 6 |

## 3. Results

### 3.1. Resistance against Subterranean Termites

SCA-treated wood samples showed increased resistance against termites with increasing WPG in both the two-choice test and the no-choice test (Figure 1). The highest WPG in SCA-treated wood resulted in nearly no mass loss and no termite attack. However, the termite survival rate in the no-choice test of SCA-treated wood samples was low even for the low treatment level (WPG 25%) during the first week and 100% after the exposure period, whereas termite survival was >50% after the two-choice and no-choice tests with untreated control wood samples (Table 3).

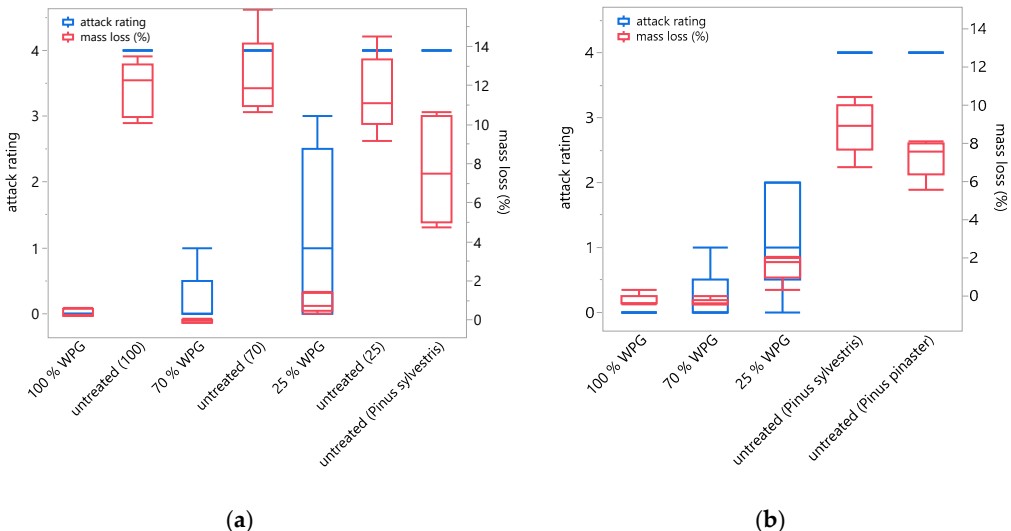

**Figure 1.** Mass loss (in red) and attack rating (in blue) of SCA-treated and untreated control wood after termite attack in (**a**) a two-choice test and (**b**) a no-choice test.

**Table 3.** Wood moisture content (MC) and termite survival rate (SR) after termite exposure in a no-choice test and a two-choice test.

| Treatment | Test Method | Final MC (%) | SR (%) |
|---|---|---|---|
| SCA100% | | 36.7 ± 6.7 | 0.0 |
| SCA70% | | 36.1 ± 5.1 | 0.0 |
| SCA25% | no-choice test | 40.1 ± 2.6 | 0.0 |
| untreated *Pinus pinaster* | | 50.0 ± 12.0 | 65.8 ± 9.8 |
| untreated *P. sylvestris* | | 106.2 ± 25.2 | 44.5 ± 15.7 |
| SCA100% | | 45.1 ± 7.7 | 58.0 ± 14.7 |
| untreated control (100) | | 90.6 ± 15.4 | |
| SCA70% | | 47.4 ± 8.0 | 67.7 ± 10.9 |
| untreated control (70) | two-choice test | 109.4 ± 44.4 | |
| SCA25% | | 61.4 ± 23.9 | 58.2 ± 24.8 |
| untreated control (25) | | 97.6 ± 39.7 | |
| untreated *P. sylvestris* | | 87.0 ± 31.4 | 64.5 ± 5.3 |

A higher survival rate was observed in the no-choice test for untreated control *P. pinaster* compared with *P. sylvestris*, which was explained by a higher presence of mold and a higher final wood moisture content of *P. sylvestris*. However, the mass loss was slightly higher for *P. sylvestris* compared with *P. pinaster*. The high attack rating on untreated control wood samples in both setups validated the test.

It was observed that termites did not avoid the SCA-treated wood material during the first days in the no-choice test. However, they died shortly after and were mostly dead within a week. A concentration threshold for SCA treatment is assumed to be between 25% and 70%.

## 3.2. Resistance against Wood Borers in the Marine Environment

The shipworm activity was high during the exposure in Moss harbor from May to September 2019 and was even slightly higher compared with the activity at the Drøbak test site, where the same test setup was used. Both test sites are located in the Oslofjord. Untreated control wood samples showed generally heavy attack (Figure 2) and untreated radiata pine wood samples even failed during the exposure (Table 4). The test is considered valid according to the standard, when all untreated control specimens have failed in less than five years. Results from the Drøbak test site, where longer experience exists compared to the Moss harbor test site, untreated control samples failed after two

years. Similar to one-year results from Drøbak, heavy attack of untreated control samples can also be seen for the Moss harbor test site.

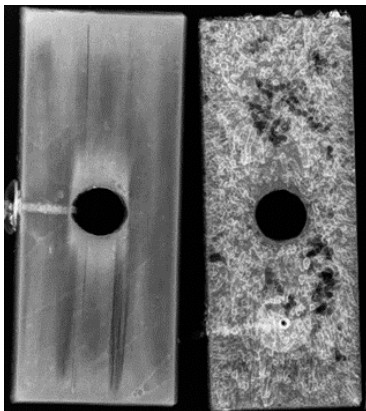

**Figure 2.** Wood borer attack on SCA-treated (**left**) and untreated control Scots pine sapwood (**right**) after four months exposure in the Oslofjord.

**Table 4.** Attack rating after 4 months of wood borer attack (shipworm) in treated and untreated control wood species in Drøbak and Moss harbor.

| Treatment | Wood Species | Test Station | N | Attack Rating (0–4) | SD |
|---|---|---|---|---|---|
| Ca oxalate | *P. sylvestris* sapwood | | 3 | 2 | 0 |
| SCA WPG25% | *P. sylvestris* sapwood | | 3 | 0 | 0 |
| SCA WPG34% | *P. sylvestris* sapwood | | 3 | 0 | 0 |
| SCA WPG50% | *P. abies* | | 2 | 0 | |
| SCA WPG50% | *P. sylvestris* sapwood | | 3 | 0 | 0 |
| SCA WPG70% | *P. abies* | | 3 | 0 | 0 |
| SCA WPG70% | *P. sylvestris* sapwood | | 3 | 0 | 0 |
| SCA WPG84% | *P. sylvestris* sapwood | Drøbak | 4 | 0 | 0 |
| SCA WPG90% | *P. abies* | | 2 | 0 | 0 |
| SCA WPG90% | *P. sylvestris* sapwood | | 3 | 0 | 0 |
| thermal modification | *F. excelsior* | | 3 | 2.6 | 2.3 |
| untreated | *P. abies* | | 3 | 2.6 | 0.6 |
| untreated | *P. sylvestris* sapwood | | 10 | 2.1 | 0.3 |
| untreated | *P. tremula* | | 1 | 2 | |
| untreated | *Q. petraea* | | 2 | 1 | 0 |
| Acetylation | *P. radiata* | | 6 | 0 | 0 |
| Ca oxalate | *P. sylvestris* sapwood | | 3 | 2.7 | 0.6 |
| SCA WPG25% | *P. abies* | | 2 | 0 | 0 |
| SCA WPG25% | *P. sylvestris* sapwood | | 3 | 0 | 0 |
| SCA WPG34% | *P. sylvestris* sapwood | | 3 | 0 | 0 |
| SCA WPG38% | *P. abies* | | 2 | 0 | 0 |
| SCA WPG50% | *P. abies* | | 2 | 0 | 0 |
| SCA WPG50% | *P. sylvestris* sapwood | | 3 | 0 | 0 |
| SCA WPG70% | *P. abies* | | 2 | 0 | 0 |
| SCA WPG70% | *P. sylvestris* sapwood | | 3 | 0 | 0 |
| SCA WPG84% | *P. sylvestris* sapwood | Moss harbor | 3 | 0 | 0 |
| SCA WPG90% | *P. abies* | | 2 | 0 | 0 |
| SCA WPG90% | *P. sylvestris* sapwood | | 3 | 0 | 0 |
| thermal modification | *F. excelsior* | | 6 | 0.2 | 0.4 |
| untreated | *E. utile* | | 6 | 0.2 | 0.4 |
| untreated | *F. sylvatica* | | 6 | 2.3 | 0.5 |
| untreated | *P. abies* | | 6 | 3 | 0 |
| untreated | *P. radiata* | | 6 | 4 | 0 |
| untreated | *P. sylvestris* sapwood | | 6 | 3 | 0.6 |
| untreated | *P. tremula* | | 6 | 3.3 | 0.8 |
| untreated | *Q. petraea* | | 6 | 1.5 | 0.5 |

Experience from the Drøbak test site in 2018 revealed a slightly lower attack rate in 2019.

The protection of wood with SCA showed no attack during the exposure period in the marine environment (Figure 2). Even the lowest treatment level, 25% WPG, of SCA-treated wood showed no attack (Table 4).

## 4. Discussion

The low survival rate of termites in the no-choice test could indicate a biocidal effect of the treatment or potential leaching from the treated samples. A high mortality can also be observed in termite tests, when biocides in a certain concentration are tested, such as permethrin or imidacloprid [33]. Studies with exposed acetylated wood showed high termite mortality rates in no-choice tests [34]. However, it is not reported if the termites died as rapidly as in this study. The number of protozoa in some termite species decreased when exposed to acetylated wood and led to starvation of the termites [35]. However, starvation of termites takes usually between one and two weeks [36]. It is therefore assumed that the mode of action of SCA treatment is incomparable to other wood modification treatments since mortality was high already within the first days of exposure. However, further studies need to include chemical analysis of the polymer in wood in order to identify the mode of action of the SCA treatment. Potential leaching of unknown substances from SCA-treated wood in decay tests has been reported and could also affect termite tests [37].

Untreated control samples in the two-choice test showed high attack and survival rate. Termites were feeding on both untreated wood samples in the control setup, preferring sometimes one side but still feeding on the other as expected as typical termite feeding behavior [38]. SCA-treated wood samples in the two-choice test were avoided by the termites, even for the lowest SCA treatment level. Termites were feeding on the untreated control sample in the same container instead and survival rate was therefore high in the two-choice setup. Water condensation on the inner side of the container was observed around the SCA-treated wood samples and to a much lower extent around untreated controls. It has been shown that the SCA treatment changes the hygroscopic behavior of wood [37,39]. These changes in wood–water relations could explain the water condensation in the container. However, the reason why termites died after a few days in no-choice tests with SCA-treated wood but survived to a large extent and avoided the treated material in two-choice tests must be investigated.

While SCA treatment showed a threshold of an effective treatment level between 25% and 70% WPG, SCA-treated wood showed no attack by marine wood borers at all treatment levels. Comparable to our study on SCA-treated wood, a correlation was found between the termite attack rate and the degree of acetylation, and a threshold at 17% WPG was indicated for acetylated wood [35].

Compared to the results from Westin et al. (2016), it is evident that one season of exposure to the marine environment is not enough to make conclusions [13]. Prolonged exposure will help to describe the service life of the SCA treatment. Other test sites, e.g., Kristineberg, Sweden, have longer exposure series and have results that show sound samples of different modified wood, such as furfurylated and acetylated wood, after 16 years [13]. Acetylated wood was included as a reference material in this study since there was no commercially available product for Use Class 5 in Europe and acetylated wood had shown good performance in other field trials. Thermally modified ash wood showed slight attack after four months of exposure in the marine environment. Good performance is not expected since heat-treated wood showed poor resistance against marine borers in a study performed at the test site in Kristineberg, Sweden [40].

The treatment of Scots pine sapwood with calcium oxalate showed severe attack. One explanation of the low protective effect of the treatment could be an insufficient reaction of the chemical components and therefore a potential leaching of unreacted chemicals [32].

However, the results of SCA-treated wood are promising since the activity of marine wood borers was high in both test sites and untreated wood samples were heavily attacked. Other test sites report a comparable service life of untreated Scots pine sapwood of one year [41].

A potential leaching of SCA during the exposure to degrading organisms, as discussed above as a possible explanation for the low survival rate of termites, is not presumed to be the case, since exposure of SCA-treated wood samples in the marine environment should have contributed to a reduced effectiveness of the treatment if the polymer is easily leached out by water.

The authors will continue their exposure test in the marine environment and provide an update on the progress in the years to come. The use of wood in the marine environment is attractive for port constructions and jetties, but also for groins and revetments, which represent two of the most challenging applications [12]. Viable and low-cost alternatives to the banned wood preservation systems in Europe need to be found if timber wants to play a bigger role in marine applications.

## 5. Conclusions

SCA-treated wood with a high treatment level showed high resistance against the tested subterranean termites in laboratory trials. The low survival rate of termites during the first week of testing in the forced feeding test and the high survival rate in the two-choice test point towards a termite-repellent effect and needs to be investigated further.

The SCA-treated spruce and Scots pine of all SCA treatment levels showed no marine borer attack after four months of exposure to the marine environment. Further studies of SCA-treated wood in the marine environment should investigate concentration thresholds, including additional marine test sites and laboratory trials and different wood deteriorating organisms, thereby providing more insight into the mode of action.

**Supplementary Materials:** The following are available online at http://www.mdpi.com/1999-4907/11/7/776/s1, Figure S1: Illustration of rating system for termite attack, Figure S2: Marine test sites in the Oslofjord, Figure S3: Illustration of rating system for marine borer attack.

**Author Contributions:** A.T. and L.N. were responsible for conceptualization, data evaluation, data validation, and formal analysis. All authors contributed to methodology. Investigations and data curation were conducted by A.T. The original draft of this article was prepared by A.T., who was also responsible for the review and editing process of this article with the help of E.L., A.T. was responsible for project administration and funding acquisition. All authors have read and agreed to the published version of the manuscript.

**Funding:** This research was funded by Norges Forskningsråd (302072) and Regionale forskningsfond Oslofjordfondet (298869).

**Conflicts of Interest:** The authors declare no conflict of interest. The funders had no role in the design of the study; in the collection, analyses, or interpretation of data; in the writing of the manuscript; or in the decision to publish the results.

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
