# Peer review of "Macrobiological Degradation of Esterified Wood with Sorbitol and Citric Acid"

_forests, doi:10.3390/f11070776_

Round 1
Reviewer 1 Report
The manuscript has been improved substantially. However, Latin names in Table 4 need to be abbreviated.
Author Response
Dear reviewer,
we appreciate your time and effort to help us improving our manuscript!
We have changed Table 4 according to your comment and abbreviated the latin names.
Reviewer 2 Report
It’s a little disappointing that the authors ignored some of the comments this reviewer provided on the first round. My opinion didn’t change. I still think the topic is interesting, but the work lacks scientific depth with respect to product characterization. I strongly suggest the manuscript be published as a Communication as opposed to a full article. I also advise the authors to strictly follow the suggestions below before the manuscript can be accepted for publication.
- The authors should mention and reference simple thermal treatment as a means to preserve wood against microorganisms (fungal degradation), as this also represents an effective, environmentally-friendly approach towards wood preservation.
- Why is moisture content calculated with m02 on the denominator? I believe this is the reason for awkward moisture contents higher than 100% in Table 3. If this is not a mistake, then the authors should explain the physical meaning of a moisture content above 100%.
Author Response
Dear reviewer,
we appreciate your time and effort to help us improving our manuscript!
As part of the group of wood modification techniques, thermal treatment can provide resistance to fungal decay. A sentence on thermal treated wood and its decay resistance is therefore added in chapter 2.3 including a new reference (Candelier et al. 2016).
However, thermal treated wood shows poor performance in service class 4 (soil contact) and when exposed to the marine environment or against termites.
See Esteves and Pereira 2009. Heat treatment of wood, BioResources 4(1), 370-404 for an overview of different thermal treatment processes and product properties.
See also Oliver-Villanueva, J., Gascón-Garrido, P. & Ibiza-Palacios, M.S. Evaluation of thermally-treated wood of beech (Fagus sylvatica L.) and ash (Fraxinus excelsior L.) against Mediterranean termites (Reticulitermes spp.). Eur. J. Wood Prod. 71, 391–393 (2013). https://doi.org/10.1007/s00107-013-0687-2 for more information on termite testing of thermal treated wood.
and Westin, M., Rapp, A., and Nilsson, T. (2006). “Field test of resistance of modified wood to marine borers,” Wood Mater. Sci. Eng.. 1(1), 34-38 for more information on the performance of thermal treated wood in the marine environment.
Other wood modification techniques, such as acetylation and furfurylation of wood, would provide higher durability in the above-mentioned use classes compared with thermally treated wood. Our manuscript focuses on these more "challenging" use classes, summarized as macro biological degradation, and offers a novel wood modification technique, namely the treatment of wood with sorbitol and citric acid.
Wood moisture content:
The calculation of wood moisture content as described in the manuscript is not a mistake, but is commonly used in the Wood Science community and described in ISO 13061-1:2014(en) Physical and mechanical properties of wood — Test methods for small clear wood specimens — Part 1: Determination of moisture content for physical and mechanical tests
or in
ASTM D4442-20, Standard Test Methods for Direct Moisture Content Measurement of Wood and Wood-Based Materials, ASTM International, West Conshohocken, PA, 2020, www.astm.org
see also chapter 12 in the Wood Handbook: https://www.fpl.fs.fed.us/documnts/fplgtr/fplgtr113/ch12.pdf
The moisture content of wood is defined as the weight of water in wood given as a percentage of oven dry weight of wood. In living trees, moisture content depends on the species and the type of wood, and may range from approximately 25% to more than 250% (two and a half times the weight of the dry wood material). In most species, the moisture content of sapwood is higher than that of heartwood. Since the water mass is related to the dry wood mass, it is possible to have wood moisture contents which exceed 100%.
This manuscript is a resubmission of an earlier submission. The following is a list of the peer review reports and author responses from that submission.
Round 1
Reviewer 1 Report
The text is well written and the topic is current and of great interest. My major criticism about the work is the lack of any characterization. All results and discussions revolve around a rating scale created by the authors and based on subjective visual observation of treated/non-treated wood specimens exposed to environment conditions. Since a visual comparison of all treated samples was not provided (this could even be presented as Supplementary Information), it’s hard to judge the correct assessment done by the authors. As is, despite being an interesting and promising idea, the work shows little scientific depth to be considered for publication. Simple analytical methods such as TGA, FT-IR, elemental analysis, XRD, microscopy imaging would greatly enhance the publication. I recommend the authors consider adding some of these analyses to better support their observations and then re-submit the manuscript. Alternatively, the manuscript could be published as a short communication. Additional suggestions are as follows:
- Maybe one point to be highlighted is the fact that citric acid and sorbitol are both bio-renewable chemicals that represent environmental advantages over non bio-based treatments commonly used for wood preservation. The authors should also mention and reference simple thermal treatment as a means to preserve wood against microorganisms (fungal degradation), as this also represents an effective, environmentally-friendly approach towards wood presenvation.
- The source of the chemicals (KCl, potassium and calcium oxalates, and CaCl2) used for chemical wood treatment need to be provided in the article.
- Table 3 – how is a moisture content of over 100% possible? The authors should explain how moisture content is calculated in the text.
- I believe the order of presentation of Figures 2 and 3 should be reversed in the text.
- The data presented in Figure 3a is repeated in Table 4. I believe Figure 3a can be omitted.
Reviewer 2 Report
Macro biological degradation of esterified wood with sorbitol and citric acid
General comments
The manuscript describes a "novel" wood protection method against marine wood borers and termites in Europe however, SCA treatment has been used to reduce susceptibility against blue stain fungi and wood decay fungi. In my opinion, the manuscript is mixing two different applications (against marine borers and termites) that do not fit in the same manuscript because in its current state both methods are very confusing for the reader. Moreover, the methodology is not carefully described and the overall work seems to be inconsistent and unclear. These might be important improvements for the wood protection field however, this should presented in more detail with more pictures and proper argumentation.
Specific comments
Page 3, L101; How the test was performed? In plastic containers or barrels? What kind of substrate was used to keep the termites active? What were the incubation conditions concerning temperature and relative humidity? How many biological replicates per container? How many biological repetitions?
Page 3, L105; What is the composition of SCA? Describe the percentage of the active substances (sorbitol and citric acid)? Is it commercially available? If so, mention the source (Company). Otherwise, describe the preparation in detail.
Page 3, L108; Latin names are only in full for the first time in the manuscript. Subsequent mentions should be abbreviated. Check throughout.
Page 3, L108; "non-choice" should be replaced by "single-choice"
Page 3, L111; There is no mention to mass loss in the results. This should be replaced by weight loss instead. Mass loss implies density and this was not analyzed here.
Page 3, 111; The conditions to record initial and final weight should also be described in detail because moister content in the wood has a big influence in the measurements. Were the samples dried in the oven for recording the measurements? If so, what was the temperature?
Page 3, 112; What was the number of termites in each container? Was consistently introduced in each box?
Page 3, 116; Abbreviations should be described for the first time (see line 105)
Page 3, 117; Citation of tables should follow numerical order in the text. Table 2 cannot be mentioned before Table 1.
Page 3, 134; Table one refers to Table 2 and this is confusing. Table 2 should be moved to the supplementary information.
Page 5, 157; If all termites were dead within a week, why did you wait 8 weeks? How did you monitor that after one week they were all dead?
Page 5, Table 3; "untreated" should be replaced by "controls". Check throughout.
Page 5, 163; How do you know that the activity was high? Are there any previous tests to compare with? Is there any threshold that EN257 states to prove efficacy in marine environments?
Page 6, 166; Location of test sites is not relevant as figures or maps. Delete. Otherwise, provide details why is this relevant.
Page 7, 175; Caption of Figure 3 is not correct. a) is showing attach effects instead of overview of the test sites. b) wood species are not providing any relevant information here nor a result
Page 7, 182; "first year" is misleading. This should be more precise. Replaced by "4 months" or "one season"
Page 7, 184; is Table 4 showing the same results than Fig. 3a)? If so, I suggest one is moved to supplementary information. Otherwise, explain the differences.
Page 8, 199; "preferring sometimes one side but still feeding on the other." This statement is very imprecise and more details should be provided. What side was preferred? Was the same pattern observed for all biological replicates?
Page 7, 215; "Longer exposure is needed to evaluate the long-term effect of the treatment.". If this was already know, why do not you wait to evaluate the real performance of your treatment? I do not imagine a wood in service that will be replaced "every season". Therefore, I do not see relevant results for the wood protection industry yet.